# The biomechanics of chewing and suckling in the infant: A potential mechanism for physiologic metopic suture closure

**Pranav N. Haravu**[1], **Miguel Gonzalez**[1], **Shelby L. Nathan**[2], **Callum F. Ross**[3], **Olga Panagiotopoulou**[4], **Russell R. Reid**[2]*

**1** Pritzker School of Medicine, University of Chicago, Illinois, United States of America, **2** Section of Plastic and Reconstructive Surgery, Department of Surgery, University of Chicago, Chicago, Illinois, **3** Department of Organismal Biology and Anatomy, University of Chicago, Chicago, Illinois, United States of America, **4** Monash Biomedicine Discovery Institute, Department of Anatomy and Developmental Biology, Monash University, Victoria, Australia

* rreid@bsd.uchicago.edu

**Data Availability Statement:** All data for this work is available at https://uchicago.box.com/s/4tvoi71ajowltdefb3jemkpwosp1v0uq.

## Abstract

Craniosynostosis is a condition with neurologic and aesthetic sequelae requiring invasive surgery. Understanding its pathobiology requires familiarity with the processes underlying physiologic suture closure. Animal studies have shown that cyclical strain from chewing and suckling influences the closure of cranial vault sutures, especially the metopic, an important locus of craniosynostosis. However, there are no human data correlating strain patterns during chewing and suckling with the physiologically early closure pattern of the metopic suture. Furthermore, differences in craniofacial morphology make it challenging to directly extrapolate animal findings to humans. Eight finite-element analysis (FEA) models were built from craniofacial computer tomography (CT) scans at varying stages of metopic suture closure, including two with isolated non-syndromic metopic craniosynostosis. Muscle forces acting on the cranium during chewing and suckling were simulated using subject-specific jaw muscle cross-sectional areas. Chewing and suckling induced tension at the metopic and sagittal sutures, and compressed the coronal, lambdoid, and squamous sutures. Relative to other cranial vault sutures, the metopic suture experienced larger magnitudes of axial strain across the suture and a lower magnitude of shear strain. Strain across the metopic suture decreased during suture closure, but other sutures were unaffected. Strain patterns along the metopic suture mirrored the anterior to posterior sequence of closure: strain magnitudes were highest at the glabella and decreased posteriorly, with minima at the nasion and the anterior fontanelle. In models of physiologic suture closure, increased degree of metopic suture closure correlated with higher maximum principal strains across the frontal bone and mid-face, a strain regime not observed in models of severe metopic craniosynostosis. In summary, our work provides human evidence that bone strain patterns from chewing and suckling correlate with the physiologically early closure pattern of the metopic suture, and that deviations from physiologic strain regimes may contribute to clinically observed craniofacial dysmorphism.

**Funding:** The authors received no specific funding for this work.

**Competing interests:** The authors have declared that no competing interests exist.

## Author summary

Developing infant skulls have multiple "sutures", which are gaps between the different bones of skull. The sutures fuse together and close at different times, and the proper sequence and timing of closure is important to allow the skull and brain to grow. If the sutures fuse together too early, children can develop a condition called craniosynostosis, which requires intensive surgical intervention to fix. However, understanding the pathologic state necessitates understanding what drives the timing of suture closure in the healthy state. There is limited human data investigating the role of mechanical forces in that process. Our work shows that chewing and suckling in infants causes unique strain patterns conducive to closure across the metopic suture (the suture that closes first in normal development), the first suture to close in normal development. Specifically, the metopic suture experienced larger amounts of compressive and tensile strain with less shearing strain i.e., sideways movement. Along the metopic suture, this strain pattern was most prominent in regions that close first, and least prominent in regions that close last. In short, our work suggests that the forces generated from chewing and suckling may be partially responsible for the timing of metopic suture closure.

## Introduction

The relative timing of cranial vault suture closure is important for normal calvarial development. Premature closure of cranial vault sutures, i.e. craniosynostosis, is a pathologic condition that affects approximately 1 in 2500 patients [1]. The premature closure results in craniofacial dysmorphism that can constrain physiologic brain growth and cause neurologic dysfunction and developmental delay [2]. The most commonly affected sutures are the sagittal and metopic sutures, and recent years have seen a rise in the frequency of metopic craniosynostosis in some reports [3–5]. In humans the metopic suture normally fuses in the first year of postnatal growth; the rapid brain growth that continues into the second year is then accommodated by appositional growth and modeling of the frontal bone [6,7]. Metopic craniosynostosis classically presents with trigonocephaly: a triangular, wedge shaped forehead with a prominent ridge along the closed metopic suture and decreased transverse frontal bone growth, as well as decreased lateral orbital rim growth [8]. Similar to other craniosynostoses, the current standard of care for treatment of metopic craniosynostosis is invasive surgery and complex reconstruction using various modalities, each carrying significant operative risks, such as cerebrospinal fluid leak, surgical site infection, and hemorrhage [9,10].

Cranial vault suture closure is hypothesized to be heavily influenced by interactions with the underlying dura mater [11,12]. The presence of the dura has been shown to be important in maintaining the patency of cranial vault sutures, and differences in cell-signaling from the dura have been shown to impact suture closure [11,12]. However, cranial vault suture closure is also thought to be influenced by mechanical stimuli such as intrauterine pressure, intracranial pressure, and chewing [13–15]. These mechanical stimuli impact suture closure at cellular level, and the application of mechanical stress, specifically cyclical compression and tension, has been shown to increase levels of osteoblast differentiation factors, decrease levels of bone morphogenetic protein (BMP) inhibitors, increase osteoprogenitor and fibroblast cell proliferation, and increase levels of extracellular matrix remodeling [13–15]. At a macroscopic level, in settings of increased compressive/tensile strain, these factors lead to suture fusion and increased interdigitation [16–18]. Interestingly, some in-vivo mice and sheep studies have also

shown that cyclic forces can augment cranial vault suture closure by maintaining patency and increasing sutural growth [19,20]. Not all types of strain are necessarily conducive to bone formation–specifically, shear strain, which may be understood as displacement that occurs when a force is applied parallel to the surface of a material, has been shown to hinder osteogenesis and fracture healing [21–24]. These data raise the possibility that mechanical stimuli may modulate suture closure in normal (physiologic) development.

One complicating factor is that the mechanical drivers—and timing of closure—vary across sutures. While the metopic closes in the first year, the remaining sutures stay patent until adulthood. This difference in closure may be partially due to differences in intrinsic factors given differences in cell lineage: only the metopic suture is completely derived from neural crest, while all other sutures are along a neural crest and mesoderm border [25,26]. However, many primate species have unfused metopic sutures as adults, and their frontal bones are presumably all derived from neural crest, suggesting that neural crest derivation alone is not a sine qua non for suture fusion [27]. Furthermore, there is evidence that the forces acting at specific cranial vault locations may supersede innate cell lineage differences in driving suture fate [28]. The importance of cranial vault location is apparent in human metopic suture closure as well, evidenced by clinically consistent patterns of closure. Specifically, the metopic suture is observed to close anterior to posterior in a 'zipper-like' fashion, beginning above the nasion and ending at the anterior fontanelle, with the nasion and anterior fontanelle closing last [29]. Given the extent to which physical stimuli have been shown to influence suture closure, differences in the mechanical landscape, specifically oscillatory tensile and compressive strain, may be partially responsible for the physiologically early closure of the metopic suture relative to other cranial vault sutures [14–17,30]. In the window of metopic suture closure, i.e. within the first year of life, the major drivers of oscillatory strain across the cranial vault sutures are likely the muscles of mastication during chewing and suckling. Animal models have shown that bilateral temporalis activation compresses the coronal and inter-frontal (i.e. metopic) sutures while tensing the sagittal suture, and that bilateral masseter activation tenses the coronal and inter-frontal sutures, but to our knowledge this has not yet been demonstrated in human infant crania, which possess different morphology than porcine and murine models [31–33].

Computational approaches such as finite element analysis (FEA) are uniquely positioned to address questions related to associations between loading regimes—external forces—acting on cranial sutures and sutural strain regimes [34–38]. FEA is widely used in biomechanical modeling, has been validated in the primate craniofacial system, and has been used to study the effects and management of craniosynostosis [39–42]. However, to our knowledge, no studies using FEA have investigated how strains resulting from the physiologic forces of chewing and suckling might impact with the closure of the metopic suture in humans.

In this study, we use patient-specific FEA to examine the role of physiologic forces, specifically muscle forces associated with chewing and suckling, in the closure of the metopic suture in humans. The animal-model based evidence leads us to hypothesize that in humans, during chewing and suckling, oscillatory compressive and tensile strains conducive to bone growth and suture closure are higher across the metopic suture than other calvarial sutures, especially at the regions of the metopic suture that are clinically observed to close first.

## Methods

### Ethics statement

This study was approved by the University of Chicago Institutional Review Board (IRB#20–2057). With IRB approval, formal consent was waived as the study used CT scans from an existing database and included patients who no longer followed at our institution and for

whom we did not have current contact information, making it unfeasible to obtain formal consent. Furthermore, radiographic data used were de-identified (anonymized) prior to analysis.

## Patient selection

Eight patient craniofacial computed tomography (CT) scans were selected from an existing database at our institution, including physiologically normal patients and patients with metopic craniosynostosis, ranging in age from 2.5 months to 3 years (Table 1). Two patients had fully patent metopic sutures, two had intermediate closure, two had physiologically closed metopic sutures, and two had isolated non-syndromic metopic craniosynostosis with trigonocephaly. Of the physiologically normal patients, 4 received CT scans due to concern for early closure of the anterior fontanelle or abnormal head shape but were found have patent sutures and fontanelles, 1 received a CT scan due to a concern for hemorrhage following trauma but was found to have no evidence of intracerebral bleeding or osseous damage, and 1 received a CT scan due to concern for a stroke secondary to a sickle cell attack but was found to have no intracerebral anomalies. The two scans with metopic craniosynostosis had interfrontal angles of 100.25° (model X1) and 104.71° (model X2) as measured from the nasion to the pterion. Each group included one male and one female patient (Table 1).

## Model creation

Patient CT scans were segmented in Mimics v21.0 (Materialise NV, Belgium) to separate cranial bone, sutures, skull base synchondroses, and the mandible. The mandible was used to calculate the force vectors during chewing and suckling, but not included in the FEA models. Paranasal sinuses, the nasal cavity, and the crypts of developing dentition were modeled as hollow spaces, per the CT scans. The synchondroses of the skull base were modeled using the same material properties as the cranial vault sutures. 3D surface datasets of the cranium and the cranial sutures were exported to 3-Matic v15.0 software (Materialise NV, Belgium) to create non-manifold files with approximately 0.6 to 2.6 million linear tetrahedral elements of 1–2 mm, and then exported to Abaqus 2021 CAE Simulia software (Dassault Systémes, Vélicy-Villacoublay, France) for modelling. Visualizations of each model to scale are provided in Fig 1, and full details of models are presented in S1 Fig. Each cranium was re-oriented into the same coordinate plane and the corresponding mandible was rotated at the temporomandibular joint (TMJ) to simulate a closed mouth position during chewing and suckling to calculate muscle vectors. Axes were oriented as follows: the (+/-) x-axis was defined as superior/inferior, the (+/-) y-axis was defined as anterior/posterior, and the (+/-) z-axis was defined as left/right mediolateral. The origin was chosen as the midpoint between the condyles of the mandible, and the horizontal x-z plane was defined as the Frankfurt plane. The interface between the

**Table 1. Summary of individuals modeled in the study.**

| Model | Metopic suture classification | Age | Sex |
|---|---|---|---|
| A1 | Fully patent | 3 mos. | F |
| A2 | Fully patent, narrowing | 3 mos. | M |
| B1 | Partially closed | 5 mos. | F |
| B2 | Partially closed | 3 mos. | M |
| C1 | Fully closed | 7 mos. | M |
| C2 | Fully closed | 3 yrs. | F |
| X1 | Isolated metopic craniosynostosis (Interfrontal angle = 100.25°) | 3 mos. | M |
| X2 | Isolated metopic craniosynostosis (Interfrontal angle = 104.71°) | 8 mos. | F |

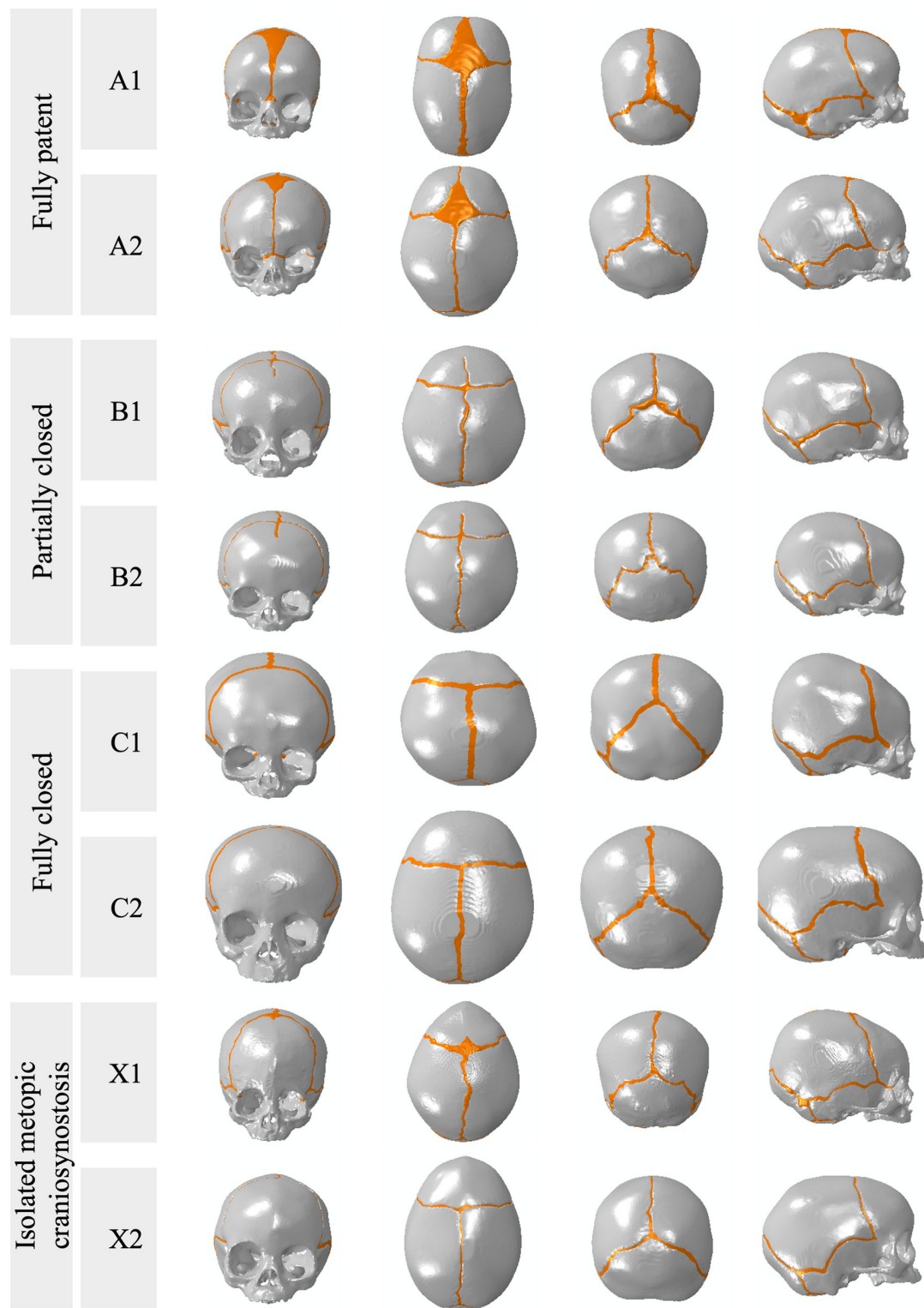

**Fig 1. Visualization of each of the models, to scale, with gray color indicating cranial bone and orange indicating cranial suture.** The views presented are, from left to right, front view, vertex view, back view, and right lateral view.

sutures and the cranial bones was modelled with tie constraints. Each model was loaded to simulate both chewing and suckling.

### Estimating muscle cross-sectional area and vectors

Muscle forces were estimated using model-specific estimates of muscle cross-sectional areas (CSA) of the deep masseter, superficial masseter, anterior temporalis, posterior temporalis, and medial pterygoid, the primary muscles acting on the cranium that are involved in mastication and suckling [43–46]. The attachment sites of the masseters, temporalis, and medial pterygoids muscles were approximated in 3-Matic based on dissections of human adult cadavers and bony landmarks on the crania and mandibles. The CSA for each muscle was estimated from the patients' CT scans. The CSA of the temporalis was measured in a transverse plane positioned immediately above and parallel to the zygomatic arch, and the CSAs of the medial pterygoid and masseter were measured in a plane at the level of the superior border of the mandibular alveolar ridge parallel to the occlusal plane [47]. Muscle CSAs were measured bilaterally and averaged for each model. The vector for each muscle was calculated as the vector from the centroid of the muscle origin on the cranium to the centroid of its insertion on the mandible. Origins and insertions for the muscles are shown in Fig 2. Details of force vectors are available in S2 Fig.

### Loading the models: chewing

Force magnitudes for each muscle were calculated by multiplying the model-specific muscle CSA with a specific tension of 22.5 N/cm$^2$ and electromyographic (EMG) activation factors

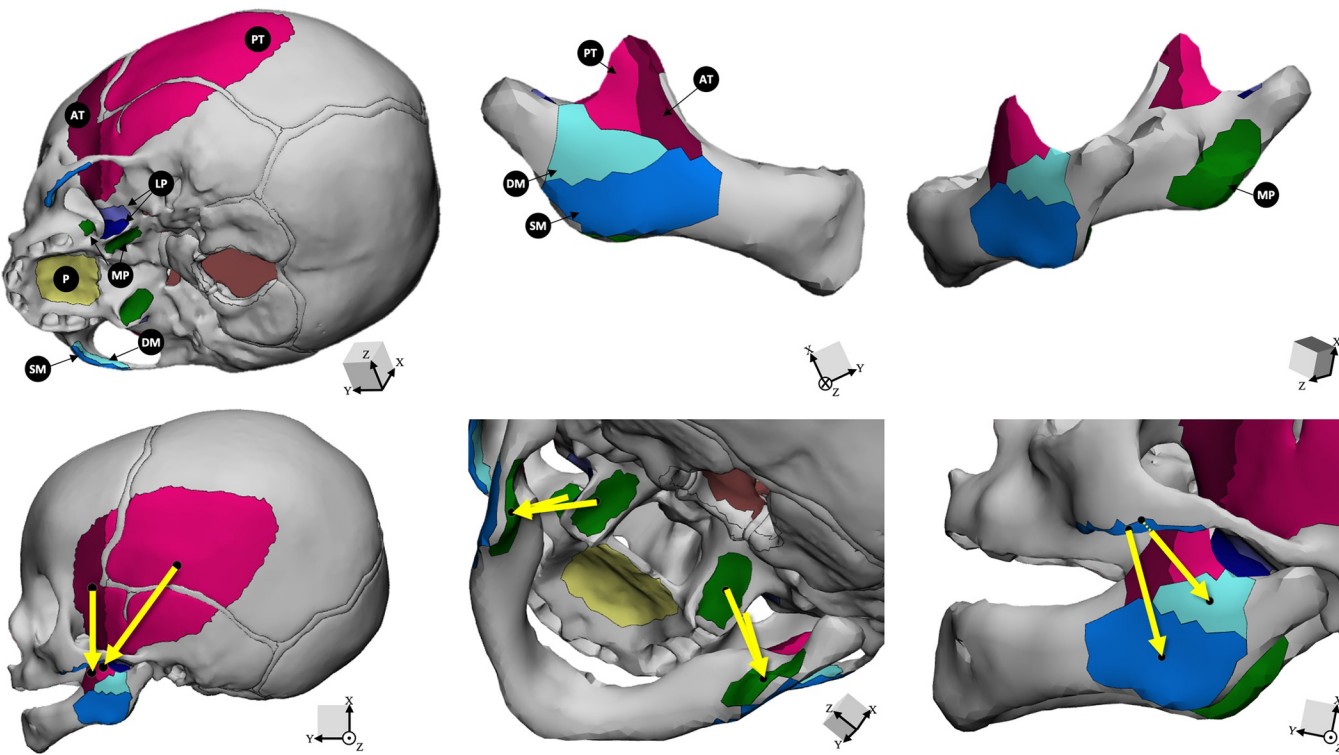

**Fig 2. Muscle attachments and force vectors for mandible and cranium.** Key: Light pink = posterior temporalis (PT); Dark pink = anterior temporalis (AT); Light blue = deep masseter (DM); Dark blue = superficial masseter (SM); Green = medial pterygoid (MP); Indigo and violet = lateral pterygoid (LP); Yellow = palate (P). Yellow arrows denote the direction of muscle vectors, from the centroid of the muscle origin on the cranium to the centroid of the muscle insertion on the mandible. Note, the LP was not included in modeling as it is not significantly involved in jaw closing but is included for ease of anatomical reference.

measured from human chewing for the superficial and deep masseter, anterior and posterior temporalis, and the medial pterygoid [44,48]. The chewing condition modeled includes bilateral muscle activation, with the right-side functioning as the working side and the left-side functioning as the balancing side. Detailed forces and vectors are available in S2 Table. Translational constraints in the $x$, $y$, and $z$ axes (no rotational constraints) were applied to the right (working side) TMJ and bite point (alveolar bone surrounding right 1$^{st}$ molar), and the left (balancing side) TMJ was constrained in the $x$ and $y$ axes [49].

## Loading the models: suckling

Force magnitudes for each muscle were calculated by multiplying the model-specific muscle CSA with a specific tension of 22.5 N/cm$^2$ and EMG activation factors measured from infant suckling during breastfeeding, with the medial pterygoid activated at the same ratio as the masseter [45]. To represent the negative intraoral pressure experienced during suckling, -87 mmHg of pressure was applied to the hard palate [50]. Detailed forces and vectors are available in S2 Table. Translational constraints in the $x$, $y$, and $z$ axes (no rotational constraints) were applied to the bilateral TMJs and midline latch point (alveolar bone between central incisors.

## Loading the models: muscle-specific and pressure-specific effects

To determine the individual contributions of each muscle group on the strain regime three loading conditions were run on the fully patent metopic suture (Model A1), each with only one of the muscle groups activated: one with bilateral deep and superficial masseter; one with bilateral anterior and posterior temporalis; and one with bilateral medial pterygoids. The muscle group was activated equally bilaterally to the working side EMG activation factor. In these individual muscle group models, boundary conditions were the same as the chewing models. The fully patent metopic suture model was also run with only the negative pressure on the palate, in which the boundary conditions were the same as the suckling models.

## Material properties

Isotropic and homogeneous material properties were assigned to the bones of the cranial vault ($E$ = 6000 MPa; v = 0.27), sutures ($E$ = 50 MPa; v = 0.30), and synchondroses of the skull ($E$ = 50 MPa; v = 0.30) based on the prior work of Jasinoski et al [51]. These material properties were used for all models and data presented in our results section.

## Model analysis

Strain (including axial, shear, and principal strain) and stress (including von Mises and Tresca) regimes across the craniofacial skeleton were analyzed both qualitatively and quantitatively. Strain across a suture (i.e. that would contribute to oscillatory compression and tension) was defined as axial strain in the axis perpendicular to the main direction of the suture. This simplification facilitates comparison of strain between sutures and is broadly acceptable, but it does not capture the impact of local deviations in suture orientation. For the nasofrontal, metopic, and sagittal sutures this was defined as medial-lateral strain ($E_{zz}$), for the coronal sutures it was defined as anterior-posterior strain ($E_{yy}$), for the squamous sutures it was defined as superior-inferior strain ($E_{xx}$), and for the lambdoid suture it was defined as the maximum of $E_{zz}$ and $E_{xx}$. These strain values, as well as the other axial and shear strains, were calculated as the average nodal strain from the surface nodes of the endo- and ectocranial surfaces of the suture. To measure strain along the midline of the cranial vault, including the

nasofrontal, metopic, anterior fontanelle, and sagittal sutures, the nodes were transformed into 2-dimensional polar coordinates about the origin, with $\theta$ defined as $arctan\left(\frac{U_x}{U_y}\right)$, with $U_x$ and $U_y$ representing the x and y-coordinates of the node respectively, depicted in S1 Fig. Cumulative shear strain was calculated as $\bar{\varepsilon}_{xy} + \bar{\varepsilon}_{xz} + \bar{\varepsilon}_{yz}$, where $\bar{\varepsilon}$ denotes the magnitude of the average of the respective nodal strain.

## Sensitivity analysis

Sensitivity analyses were conducted using the patent metopic suture model (A1) during simulated chewing. The impact of material properties on model behavior was assessed first. Two additional sets of cranial bone material properties based on the work of Borghi et al. (bone, $E$ = 418 MPa) and Barbeito-Andres et al. (bone, $E$ = 1300 MPa) were simulated and resulting maximum principal strain regimes were compared to the results that utilized material properties from Jasinoski et al. (bone, $E$ = 6000 MPa), as detailed in S2 Fig [52,53]. While decreased $E$ predictably resulted in higher strain magnitudes, the pattern and relative values of strain remained similar across models. Next considered were the utilized boundary conditions. FEA models focused on calvarial growth often constrain translation at the foramen magnum, but this is not a biologically realistic constraint [39]. Sensitivity analyses of the FEA model run with and without the foramen magnum constraint (S3 Fig) depicts the minimal differences between the two set-ups. Given the similarity, the unconstrained option was chosen to reduce the likelihood of over constraining the model. Last, variations in bite point were considered (S4 Fig), but the impact of varying the bite point was largely limited to strain differences in the nearby areas of the alveolar bone, nasal walls, and orbital floor.

## Results

### Understanding the models: muscle specific effects

In order to understand the results of the chewing and suckling FEA models, it is important to understand the individual impact of each force on the models. The impact of individual muscle groups and negative palatal pressure on the strain regimes in the model with a fully patent metopic suture (A1) are depicted in Fig 3 and Table 2. A summarized schematic of the axial compressive and tensile strains across the sutures is depicted in Fig 4. Bilateral activation of the temporalis compressed the squamous sutures along the superior-inferior axis (negative $\varepsilon_{xx}$), placed them under positive sagittal shear strain ($\varepsilon_{xy}$), and subjected them to high magnitudes of coronal shear strain ($\varepsilon_{xz}$). The temporalis also placed all sutures under medial-lateral tension (positive $\varepsilon_{zz}$) and resulted in compressive anterior-posterior strain (negative $\varepsilon_{yy}$) on the anterior fontanelle, coronal suture, and metopic suture. The temporalis, masseter, and medial pterygoid muscles caused negative coronal shearing (negative $\varepsilon_{xz}$) in the nasofrontal suture. The masseter also tensed the metopic suture, sagittal suture, nasofrontal suture, and anterior fontanelle in the medial-lateral axis. On the balancing (left) side the masseter compressed the squamous suture, while tensing it on the working (right) side. The medial pterygoid compressed (negative $\varepsilon_{zz}$) and tensed (positve $\varepsilon_{yy}$) the nasofrontal suture along the medial-lateral and anterior-posterior axes respectively and compressed the metopic suture along the medial-lateral axis. The magnitude of strain arising from the impact of the negative palatal pressure was lower in magnitude than the strain from the activation of the masticatory muscles. It resulted in the squamous suture experiencing superior-inferior tension with positive sagittal shear, the nasofrontal suture experiencing medial-lateral tension, and the coronal suture experiencing anterior-posterior tension and positive sagittal shear.

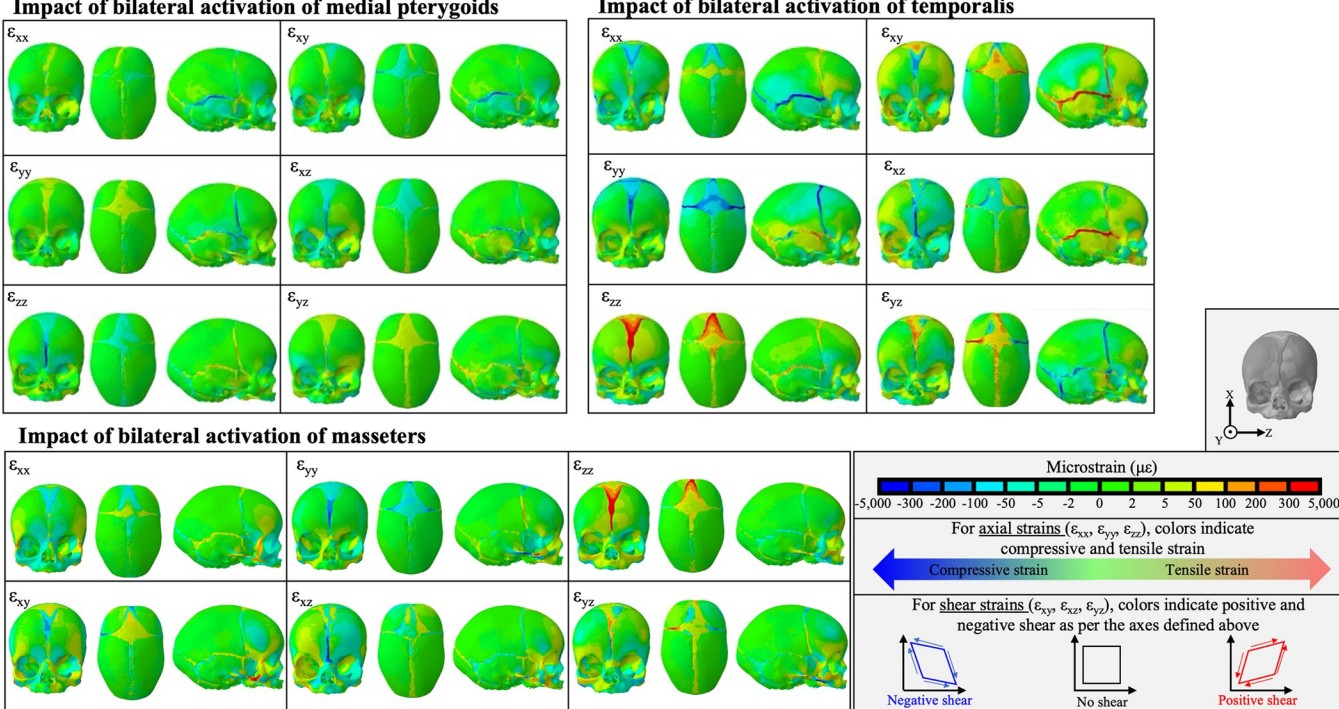

**Fig 3. Strain regimes created by bilateral activation of each of the major masticatory muscle groups during right sided chew in model A1, with a fully patent metopic suture.** Axial strains denoted by $\varepsilon_{xx}$ (superior-inferior), $\varepsilon_{yy}$ (anterior-posterior), and $\varepsilon_{zz}$ (medial-lateral). Shear strains denoted by $\varepsilon_{xy}$ (sagittal shear), $\varepsilon_{xz}$ (coronal shear), and $\varepsilon_{yz}$ (transverse shear). For axial strains, warm colors indicate tension and cool colors indicate compression. For shear strains, warm colors denote positive shear and cool colors denote negative shear. The medial pterygoids placed the metopic suture under medial-lateral compression, the coronal sutures under anterior-posterior compression, and the squamous sutures under superior inferior compression. The temporalis placed those sutures under tension, compression, and compression respectively. The temporalis also placed large amounts of shear strain on all the cranial vault sutures. The masseters placed the metopic suture under medial-lateral tension and the nasofrontal suture under negative transverse shear.

## Understanding the models: stress and strain regimes in the mid face

Axial stresses and strains over the mid-face during chewing are shown in Fig 5. Stress and strain regimes in the midfacial skeleton showed similar patterns in all models regardless of the degree of metopic suture closure and were similar in both chewing and suckling behaviors. Superior-inferior stress ($\sigma_{xx}$) was concentrated along the vertical maxillofacial buttresses, specifically the lateral maxillary buttress [54]. This resulted in tensile (e.g. positive) superior-inferior strain ($\varepsilon_{xx}$) along the lateral maxillary buttress from the inferior aspect of the zygoma through the lateral orbital wall with concomitant compressive (e.g. negative) $\varepsilon_{xx}$ along the nasomaxillary buttress from the canine fossa to the maxillary frontal process along the medial orbital wall. Medial-lateral stress ($\sigma_{zz}$) also followed the maxillary buttresses and was highest along the lower transverse buttress along the alveolar process and the upper transverse buttress along the inferior orbital rim. This corresponded to tensile medial-lateral strain ($\varepsilon_{zz}$) along the upper transverse maxillary buttress and compressive $\varepsilon_{zz}$ along the lower transverse maxillary buttress. Similar stress and strain regimes were observed during suckling and are depicted in S5 Fig.

## Strains across sutures

When analyzing the axial strains perpendicular to the suture, the metopic suture and sagittal suture experienced tensile strains during chewing and suckling behaviors, while the coronal, lambdoid, and squamous sutures experienced compressive strains (Fig 6). A summarized

**Table 2. Average strain at the sutures from the bilateral activation of the respective muscle groups and the application of negative palatal pressure.**

| Model | Suture | Average microstrain (µε) | | | | | |
|---|---|---|---|---|---|---|---|
| | | $\varepsilon_{xx}$ | $\varepsilon_{yy}$ | $\varepsilon_{zz}$ | $\varepsilon_{xy}$ | $\varepsilon_{xz}$ | $\varepsilon_{yz}$ |
| Temporalis | Anterior fontanelle | -7.1 | -100.2 | 101.2 | 70.0 | -16.5 | 60.2 |
| | Coronal (L, BS) | 37.9 | -322.8 | 31.7 | -63.6 | 64.7 | 340.5 |
| | Coronal (R, WS) | 63.0 | -330.2 | 50.0 | 179.6 | -7.3 | -185.7 |
| | Lambdoid (L, BS) | -65.0 | 2.0 | 50.2 | 139.5 | 21.2 | 82.6 |
| | Lambdoid (R, WS) | -90.9 | 55.3 | 22.4 | 57.5 | 36.8 | -84.4 |
| | Metopic | -82.7 | -104.6 | 375.8 | -24.1 | -51.7 | 81.0 |
| | Nasofrontal | -3.2 | -1.1 | 7.7 | 7.5 | -272.0 | -64.6 |
| | Posterior fontanelle | -29.5 | -3.0 | 31.7 | -20.5 | 49.3 | 17.8 |
| | Sagittal | -16.0 | -9.8 | 54.3 | 3.4 | 29.7 | 81.5 |
| | Squamous (L, BS) | -679.4 | 5.0 | 178.3 | 220.2 | -493.8 | 56.9 |
| | Squamous (R, WS) | -360.9 | 16.2 | 70.9 | 393.6 | 408.8 | -102.8 |
| Masseter | Anterior fontanelle | -6.0 | -43.8 | 51.9 | 27.5 | -2.3 | 10.9 |
| | Coronal (L, BS) | -7.7 | 2.8 | -9.3 | -132.3 | 46.1 | 70.3 |
| | Coronal (R, WS) | 7.7 | -15.7 | -4.6 | -2.4 | -8.2 | -31.4 |
| | Lambdoid (L, BS) | 1.4 | -2.8 | 7.1 | 12.0 | 14.8 | 11.4 |
| | Lambdoid (R, WS) | -8.6 | 7.6 | 1.0 | -6.0 | 5.1 | -7.1 |
| | Metopic | -54.6 | -75.5 | 274.5 | -25.6 | -15.7 | 26.4 |
| | Nasofrontal | -15.7 | -11.6 | 61.7 | 16.4 | -259.0 | -56.8 |
| | Posterior fontanelle | -5.4 | -0.4 | 5.5 | -3.7 | 13.1 | 4.9 |
| | Sagittal | -5.4 | -3.6 | 17.9 | 0.0 | 8.6 | 24.1 |
| | Squamous (L, BS) | -74.3 | -6.2 | 29.2 | -20.8 | -23.9 | -6.1 |
| | Squamous (R, WS) | 19.9 | -2.1 | -6.3 | 5.6 | -5.2 | -5.1 |
| Medial pterygoid | Anterior fontanelle | -0.6 | 10.9 | -7.8 | -8.4 | -10.6 | 32.5 |
| | Coronal (L, BS) | -23.4 | -32.1 | 21.4 | -171.4 | 51.6 | 28.9 |
| | Coronal (R, WS) | -8.2 | -68.7 | 29.8 | -51.3 | -10.7 | 3.4 |
| | Lambdoid (L, BS) | 23.2 | -22.3 | -7.3 | -19.9 | 1.1 | -44.2 |
| | Lambdoid (R, WS) | 17.4 | -12.5 | -15.9 | -52.9 | 27.9 | 52.4 |
| | Metopic | 5.1 | 14.1 | -45.3 | -1.6 | -47.5 | 30.3 |
| | Nasofrontal | -2.7 | 41.2 | -150.6 | -21.6 | -118.8 | -21.7 |
| | Posterior fontanelle | 11.8 | 0.9 | -11.8 | 8.3 | 16.8 | 7.9 |
| | Sagittal | 2.4 | 1.4 | -7.9 | -0.2 | 5.4 | 30.0 |
| | Squamous (L, BS) | -126.6 | -7.4 | 56.1 | -131.8 | -1.2 | -58.9 |
| | Squamous (R, WS) | -85.3 | 7.2 | 30.8 | -197.4 | -30.4 | 38.5 |
| Negative palatal pressure | Anterior fontanelle | 0.1 | -0.1 | -0.2 | 0.3 | 0.0 | -0.2 |
| | Coronal (L) | 1.1 | 3.7 | -1.7 | 8.6 | -3.0 | -1.6 |
| | Coronal (R) | 1.0 | 4.2 | -1.7 | 8.0 | 2.0 | 1.9 |
| | Lambdoid (L) | -1.2 | 1.2 | 0.7 | 2.0 | 0.7 | 3.3 |
| | Lambdoid (R) | -1.3 | 1.2 | 0.7 | 2.4 | -1.1 | -3.3 |
| | Metopic | 0.2 | -0.1 | 0.2 | 0.1 | 1.1 | -0.6 |
| | Nasofrontal | 0.4 | -2.8 | 9.6 | 1.1 | 3.6 | 0.7 |
| | Posterior fontanelle | -0.8 | 0.0 | 0.7 | -0.6 | -0.1 | -0.1 |
| | Sagittal | -0.1 | -0.1 | 0.2 | 0.0 | 0.0 | -0.1 |
| | Squamous (L) | 6.6 | 0.3 | -3.0 | 8.7 | -0.4 | 3.7 |
| | Squamous (R) | 9.3 | -0.5 | -3.3 | 13.5 | 1.0 | -2.8 |

Abbr: $\varepsilon_{xx}$ is superior-inferior axial strain, $\varepsilon_{yy}$ is anterior-posterior axial strain, and $\varepsilon_{zz}$ is medial-lateral axial strain. $\varepsilon_{xy}$ = sagittal shear strain; $\varepsilon_{xz}$ = coronal shear strain; $\varepsilon_{yz}$ = transverse shear strain. L, R indicate left and right respectively. BS, WS indicate balancing and working sides respectively. There is no distinction between working and balancing side in the negative palatal pressure model given the midline latch point. Values color coded with red (+, tensile strain) and blue (-, compressive strain) for axial strains; red (+, positive shear strain) and blue (-, negative shear strain) for shear strains.

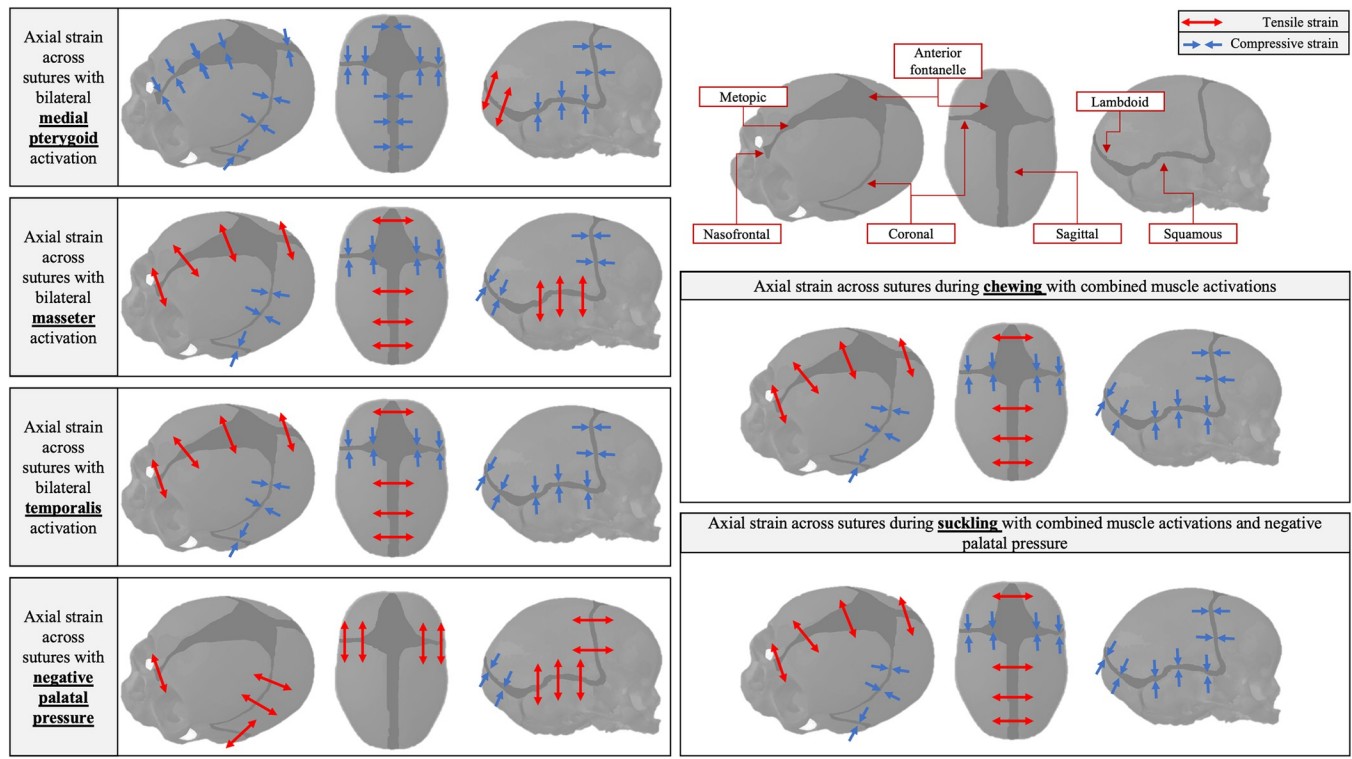

**Fig 4. Direction of axial strain across cranial vault sutures during bilateral activation of individual muscle groups (medial pterygoid, masseter, temporalis) with a right-sided chew, applied negative palatal pressure with a midline latch point, physiologic chewing, and physiologic suckling.** Cranial vault sutures are labeled in top right panel, alongside the legend for tensile (red arrows) and compressive (blue arrows) strain. Note, arrows only denote direction of axial strain across the suture (i.e., compression or tension), not the magnitude. No arrows are included for axial strain across the metopic suture, anterior fontanelle, and sagittal suture in the negative palatal pressure model due to the low magnitude of strain across these sutures (<1 με).

schematic depicting the direction (compressive vs. tensile) of the axial strain across the sutures during chewing and suckling is presented in Fig 4. When considering principal strains, the metopic suture was predominantly under tension, evidenced by larger magnitudes of maximum principal strain ($\varepsilon_1$) than minimum principal strains ($\varepsilon_3$). Average $\varepsilon_1$ along the metopic suture during chewing and suckling in model A1 was 527 με and 763 με respectively, while $\varepsilon_3$ was -200 με and -291 με respectively. In model A2, average $\varepsilon_1$ along the metopic suture during chewing and suckling was 332 με and 333 με, while $\varepsilon_3$ was -162 με and -167 με respectively. In models with a fully patent metopic suture (A1 and A2), the metopic suture experienced higher magnitudes of compressive/tensile strains during both chewing and suckling than all other cranial vault sutures except the squamous suture. During chewing the metopic suture was under median strains of 300–475 με when fully patent and 100–125 με when partially closed. During suckling it was under 375–650 με when fully patent and 125–250 με when partially closed. Between the two models with a fully patent metopic suture, the model with the narrower metopic suture (A2) tended to have a lower magnitude of axial and shear strain across the suture, in line with the trend of lower strain magnitudes across and along the suture as it progresses towards closure. Notably, the squamous and coronal sutures experienced larger magnitudes of shear strain than the metopic, sagittal, and lambdoid sutures (Fig 7). In the fully patent models the range of absolute magnitude of shear strain in the metopic suture was within 500 με, but the magnitude of shear strain in the coronal and squamous sutures surpassed 1000 με and 1500 με respectively. The degree of metopic suture closure (patent vs. partial vs. fully closed) had no discernible impact on compressive/tensile strains across the remaining

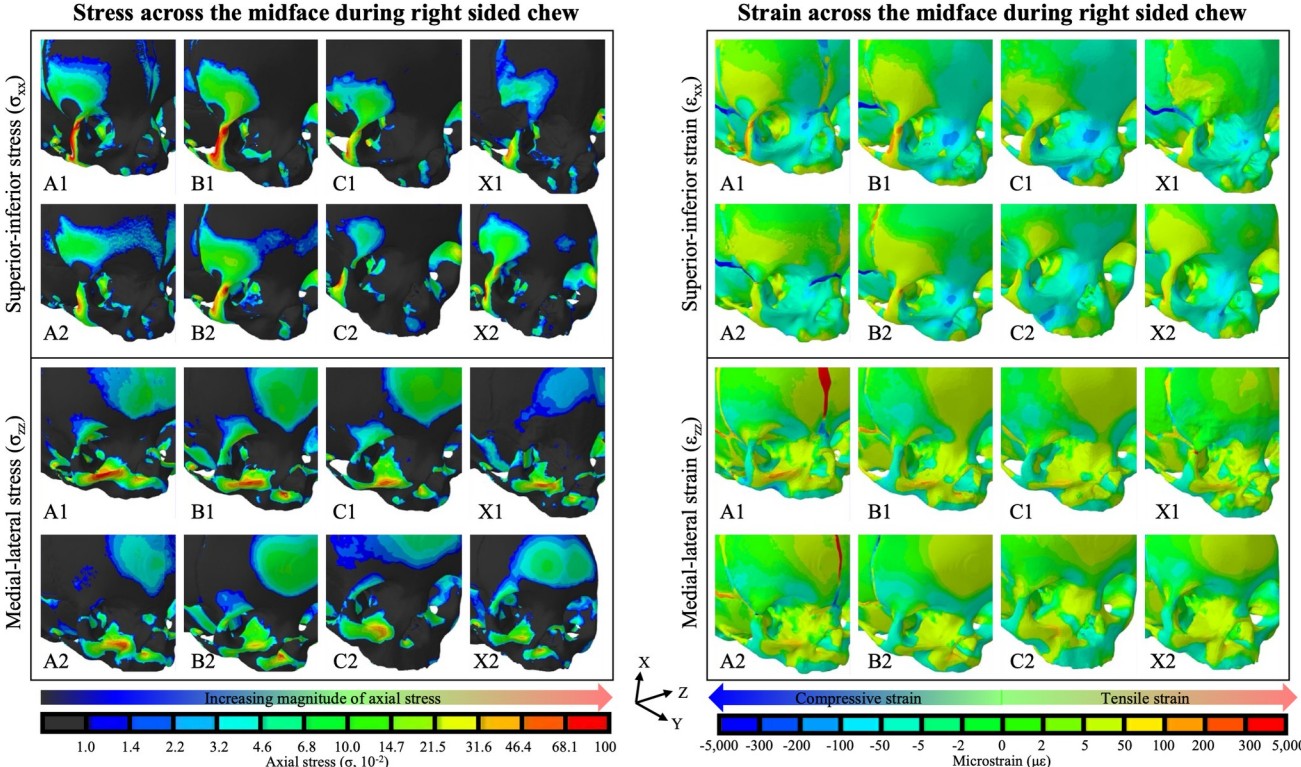

**Fig 5. Axial stress and strain in the superior-inferior ($\sigma_{xx}$ and $\varepsilon_{xx}$) and medial-lateral ($\sigma_{zz}$ and $\varepsilon_{zz}$) directions in each of the models during right-sided chew.** Left and right panels respectively shown stress and strain. In all models, superior-inferior stress and strain were the highest along the lateral maxillary vertical buttress, from the zygoma along the lateral orbital wall. The high levels of superior-inferior axial stress along the lateral orbital wall corresponded to tensile (positive, red) superior-inferior strain in the region, with corresponding compressive (negative, blue) superior-inferior strain along the medial orbital wall. Medial-lateral stress and strain were the highest along the transverse maxillary buttress, specifically the lower transverse buttress along the alveolar process and the upper transverse buttress along the inferior orbital rim. Stress and strain following the maxillary buttresses provides supporting evidence that the models are correctly modeling the physiologic mechanical landscape.

patent cranial vault sutures during either chewing or suckling. Direct strains across patent cranial vault sutures in the trigonocephalic metopic craniosynostosis models were similar to those in the fully closed physiologic model.

## Strain across the mid-sagittal plane of the cranial vault

During both chewing and suckling, sutures along the mid-sagittal plane of the cranial vault showed maximum and minimum values of compressive/tensile (e.g. medial-lateral $\varepsilon_{zz}$) strain that correlated with the clinically observed pattern of suture closure (Fig 8). When fully patent (model A1), the metopic suture had maximal $\varepsilon_{zz}$ strain at the most inferior end, proximal to the nasion, which then decreased along the suture in the direction of the anterior fontanelle. Both ends of the metopic suture, at the nasion and the anterior fontanelle, experienced local $\varepsilon_{zz}$ minima. In the intermediate closure model (model A2), where the metopic suture was still patent yet narrower than A1, the local $\varepsilon_{zz}$ minima at the nasion and anterior fontanelle were still observed but the $\varepsilon_{zz}$ strain pattern along the metopic suture did not show a prominent local maximum at the inferior end proximal to the nasion. Relative to the fully patent A1 model, $\varepsilon_{zz}$ strain magnitudes along the A2 model tended to be lower in magnitude, consistent with the observed trend of decreased strain magnitudes along and across the metopic suture as

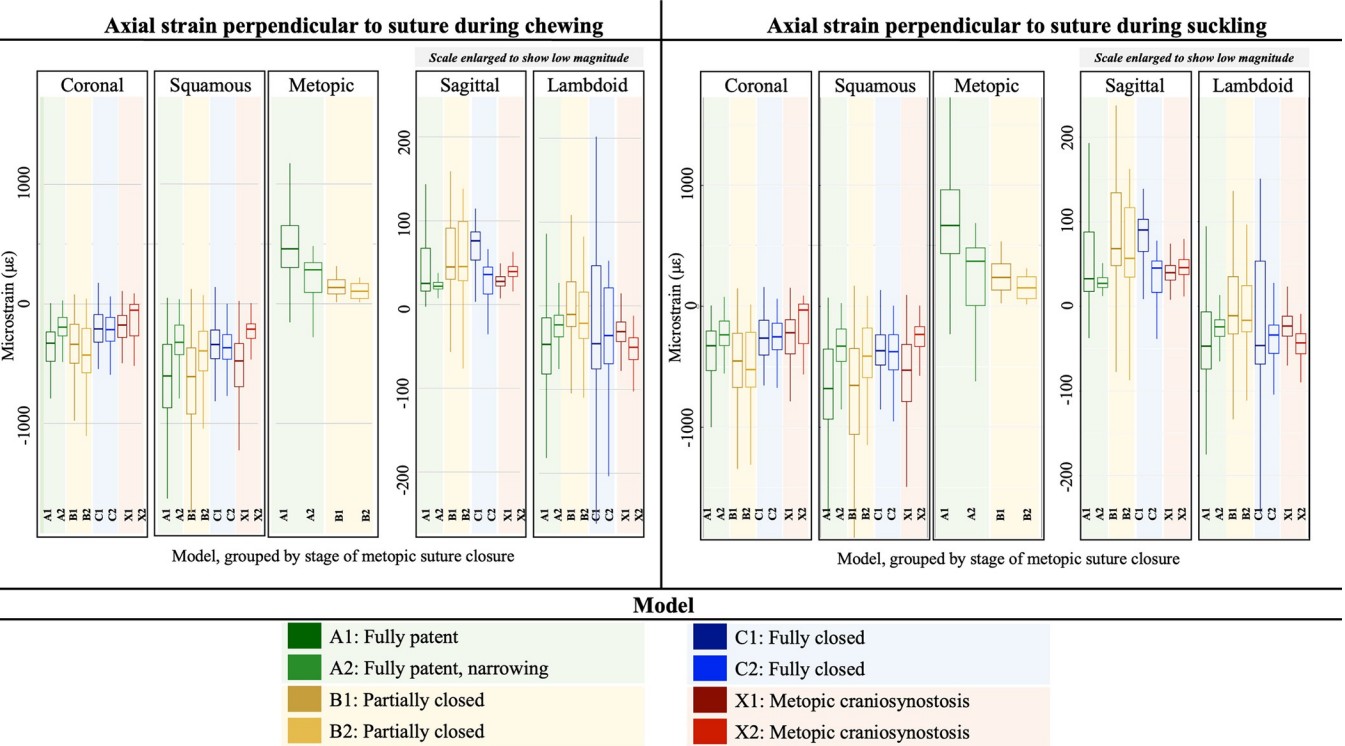

**Fig 6.** Strains perpendicular to suture for each model during chewing (Left) and suckling (Right), box-plots color coded by stage of metopic suture closure. Green plots are models with fully patent metopic suture, yellow plots are models with partially closed metopic sutures, blue plots are models with physiologically fully closed metopic sutures, and red plots are models with metopic craniosynostosis. Positive and negative strain respectively show tension and compression. For sagittal and metopic sutures, axial strain perpendicular to the suture is medial-lateral strain; coronal suture = antero-posterior strain; lambdoid suture = max of superior-inferior and medial-lateral strains; squamous = superior-inferior strain. Models with fully patent metopic sutures had highest magnitudes of cross-sutural strain across the metopic and squamous sutures. In models with increasing degrees of metopic suture closure, the magnitude of cross-sutural strain across the metopic suture decreased, but no such trend was observed across the remaining sutures. Cross-sutural strain during suckling tended to be of higher magnitudes than during chewing.

it progressed towards closure. In models B1 and B2, which showed partial closure of the metopic suture, the local $\varepsilon_{zz}$ minimum at the anterior fontanelle was still present, and the $\varepsilon_{zz}$ strain pattern along the metopic suture showed an inverted pattern relative to model A1, with medial-lateral strain increasing towards the anterior fontanelle. In models with closed metopic sutures (C1, C2, X1, and X2), $\varepsilon_{zz}$ strain across the sagittal suture was within a range of 50 μɛ from the anterior fontanelle to the posterior fontanelle with the exception of a local minimum experienced at the anterior sagittal suture.

In the fully patent models (A1, A2), cumulative shear strain at the metopic suture was 1.0x – 3.0x higher during chewing than suckling. Local maxima of shear strain were observed at the anterior fontanelle and bregma in partially closed (B1, B2) and fully closed (C1, C2, X1, X2) models. The magnitudes of shear strain along the mid-sagittal plane of the cranial vault reached local maxima at the nasion and anterior fontanelle during both chewing and suckling (Fig 9). However, in models with partially closed metopic sutures (B1, B2), the shear strain at the anterior fontanelle was greater than the shear strains at the metopic suture.

No difference was noted in the strain pattern along the midline of the cranial vault between the trigonocephalic models (X1, X2) and the fully closed physiologic models (C1, C2). No difference in strain values was computed between the endocranial and ectocranial surfaces of the metopic suture.

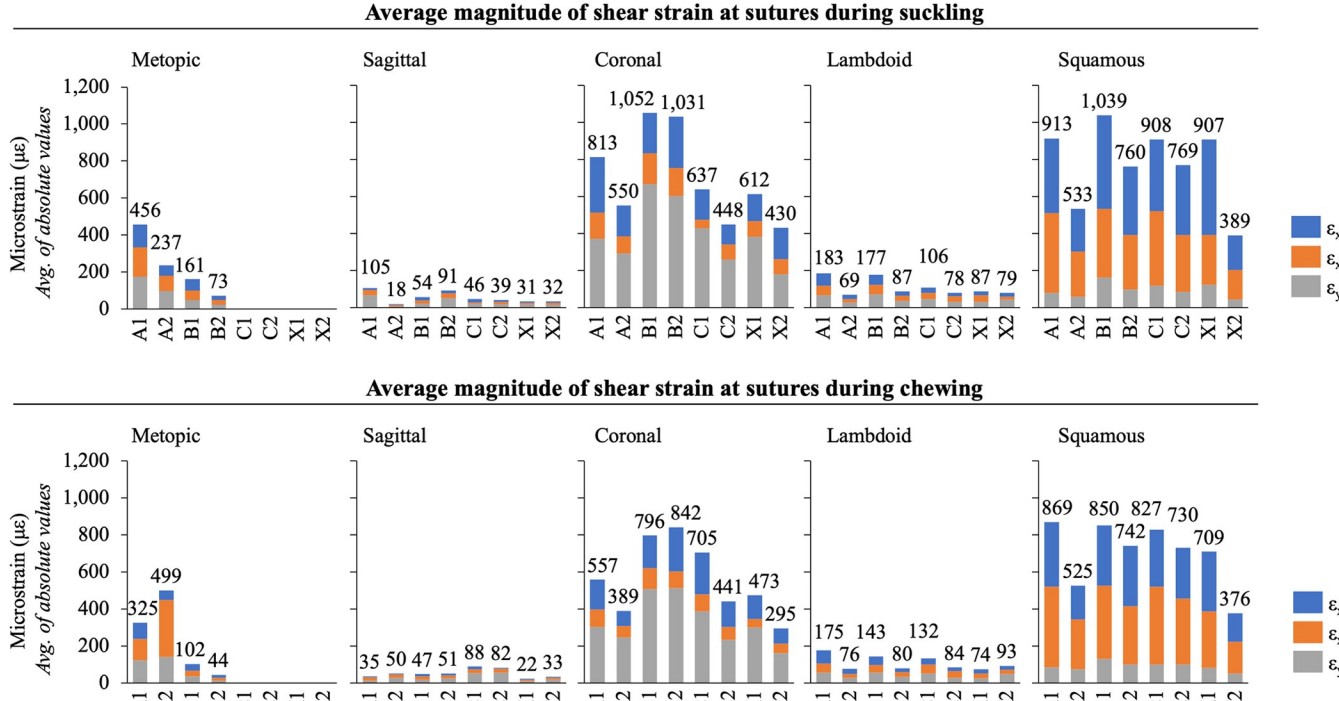

**Fig 7.** Average shear **st**rains experienced by each suture in each model during suckling (top panel) and chewing (bottom panel). Grey bar ($\varepsilon_{yz}$) indicates average magnitude of transverse shear, orange bar ($\varepsilon_{xz}$) indicates average magnitude of coronal shear, and blue bar ($\varepsilon_{xy}$) indicates average magnitude of sagittal shear. The number atop each bar represents the sum of each of the average magnitudes. The largest shear strains were experienced by the squamous and coronal sutures. In models with a fully patent metopic suture (A1, A2), and in models with a partially closed metopic suture (B1, B2), the total shear strain experienced by the coronal and squamous sutures tended to be higher than the total shear strain experienced by the metopic suture.

## Strains in the upper face

In addition to the primary hypotheses examining the cranial vault sutures, differences in strain values across the frontal and orbital cranium were also observed. For physiologic models (e.g. non-metopic craniosynostosis), the frontal eminence experienced increasing levels of strain with increased degree of metopic suture closure (Fig 10). When fully patent (models A1, A2), during chewing and suckling, maximum principal strains throughout the frontal bone were mostly < 20 με. In models with partially closed (models B1, B2) and physiologically closed metopic sutures (models C1, C2), most of the frontal eminence experienced maximum principal strains of 20–50 με with hotspots up to 70 με. In the more severe metopic craniosynostosis (model X1) maximum principal strains in the supraorbital regions and frontal eminences were <20 uE during both chewing and suckling.

The von Mises stress regimes during chewing and suckling followed a similar trend. The frontal prominence, supraorbital region, and lateral orbital wall all experienced lower stresses in the severe metopic craniosynostosis model (X1) when compared to the non-metopic cranio-synostosis, i.e. physiologically closed, models (C1, C2) (Fig 11). The less severe metopic cranio-synostosis model (X2) had strain and stress regimes that mirrored that of the physiologically closed metopic suture models.

## Discussion

Our results show that during physiologic chewing and suckling in humans, the patent metopic suture experiences strain regimes that may correlate with its unique physiologic fate, i.e. early

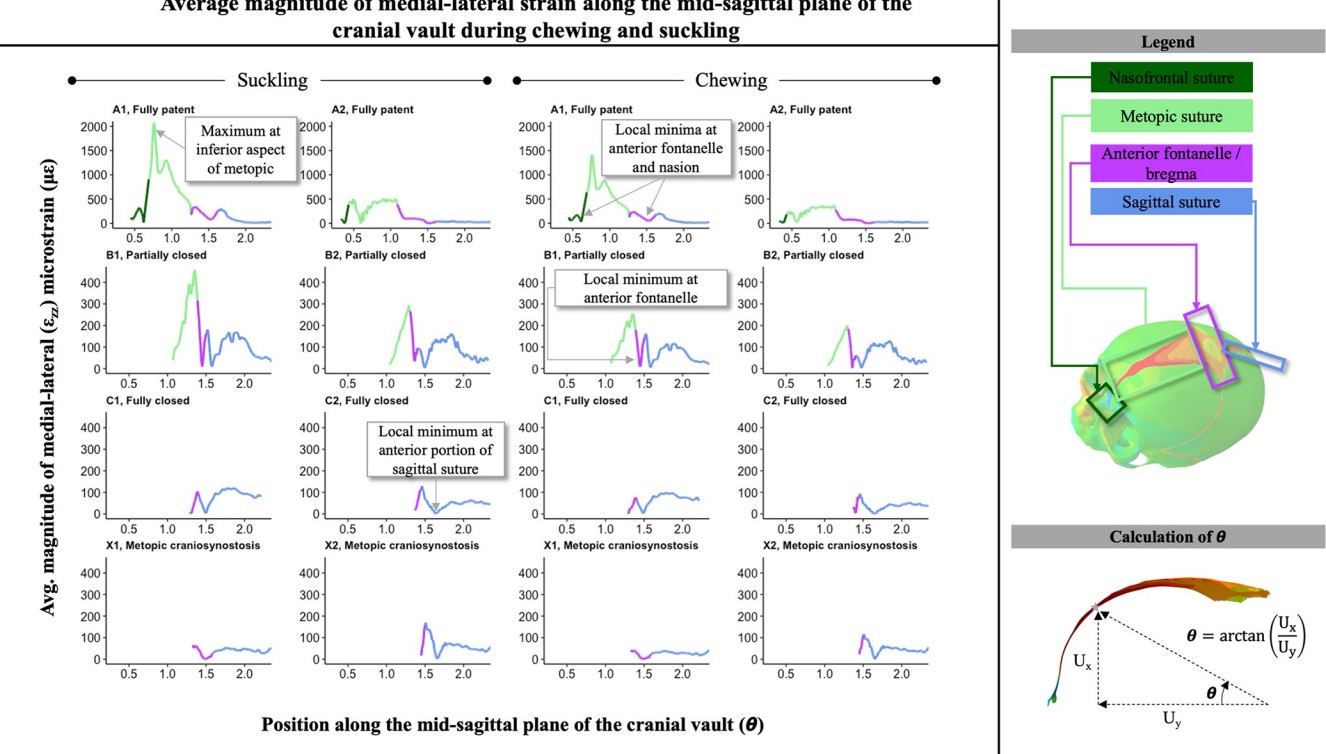

**Fig 8. Average magnitude of medial-lateral strain ($\varepsilon_{zz}$) along mid-sagittal plane during right-sided chew and suckling for models in the respectively labeled stages of metopic suture closure.** Theta calculated as $arctan\left(\frac{U_x}{U_y}\right)$, with $U_x$ and $U_y$ representing the x and y-coordinates of the node respectively; moving left to right along the x-axis of each graph is the path from the nasion along the metopic suture, past the anterior fontanelle, and along the sagittal suture towards the posterior fontanelle. Color coding represents which portion of the mid-line of the cranial vault is being represented. In both fully patent metopic models (A1, A2), there were local minima of compressive/tensile medial-lateral strain at the anterior fontanelle and nasion. In A1, the fully patent metopic suture model in which the metopic suture had not yet begun narrowing, a maximum strain value was observed at the inferior aspect of the metopic suture. In the partially closed models (B1, B2), local minima were observed at the anterior fontanelles.

closure. Specifically, when compared to the other cranial vault sutures, the patent metopic suture experiences higher magnitudes of axial medial-lateral strain across the suture with minimal shear strain, especially at the areas of the metopic suture that close first (Figs 4 to 7). Within the fully patent metopic suture, there is a local maximum of compressive/tensile strain superior to the nasion that decreases to a local minimum at the anterior fontanelle (Figs 6 and 7). This correlates with the clinically observed pattern of metopic suture closure: anterior to posterior in a 'zipper-like' fashion, beginning above the nasion and ending at the anterior fontanelle [29]. The last two areas to close, the nasion and the anterior fontanelle, experienced minimum values of compressive/tensile strain and local maximums of shear strain. As the patent metopic suture begins to close, axial strain across it begins to decrease, while axial strains across the remaining cranial vault sutures remain unchanged. The fact that the other cranial vault sutures remain patent well into adulthood, combined with the unique regional axial strains our models have identified specific to the metopic region, enhance the validation of our analyses. Taken in combination, these results suggest that the physiologic forces of chewing and suckling result in strain regimes that map closely onto the timing of physiologic metopic suture closure relative to other cranial vault sutures, as well as the observed pattern of closure along the suture.

Studies of animal models support this hypothesis: during mastication in pigs the interfrontal (i.e. equivalent to human metopic) suture experiences larger strain magnitudes than other

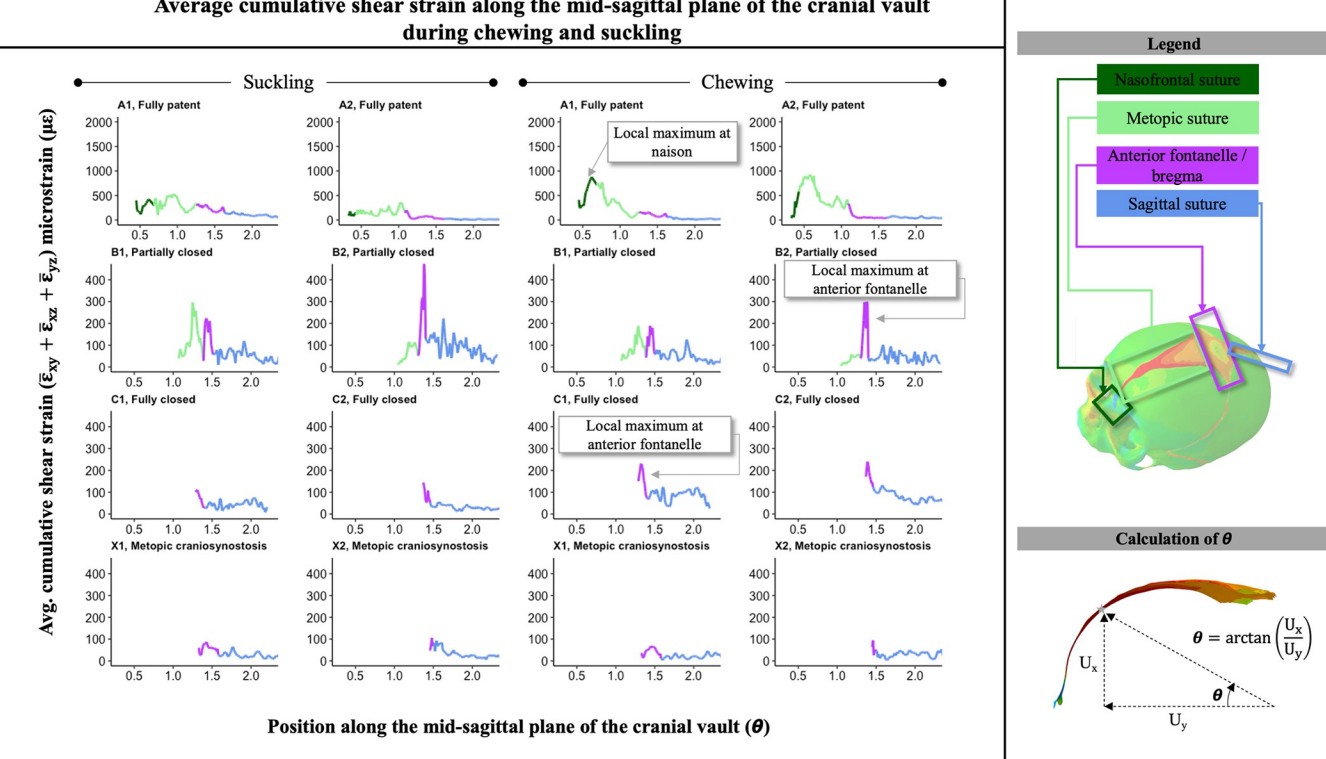

**Fig 9. Cumulative shear strain ($\bar{\varepsilon}_{xy} + \bar{\varepsilon}_{xz} + \bar{\varepsilon}_{yz}$, where $\bar{\varepsilon}$ denotes the magnitude of the average of the respective nodal strain) along mid-sagittal plane during right-sided chew and suckling for models in the respectively labeled stages of metopic suture closure.** Theta calculated as $arctan\left(\frac{U_x}{U_y}\right)$, with $U_x$ and $U_y$ representing the x and y-coordinates of the node respectively; moving left to right along the x-axis of each graph is the path from the nasion along the metopic suture, past the anterior fontanelle, and along the sagittal suture towards the posterior fontanelle. Color coding represents which portion of the midline of the cranial vault is being represented. In fully patent models (A1, A2) cumulative shear strain was higher during chewing than suckling at the metopic suture, and in A1, a local maximum was observed at the nasion. In partially and fully closed models (B1, B2, C1, C2, X1, X2), local maxima were observed at the anterior fontanelle.

cranial vault sutures [32,55]. However, caution is warranted when using findings in animal models to interpret the situation in humans. First, there is evidence that cranial vault suture closure is evolutionarily conserved, with increasing differences the further one gets in taxonomy from humans [56,57]. For example, in mice, the interfrontal suture closes shortly after weaning but other sutures remain patent throughout life [58]. In addition, there are significant morphological differences between human craniofacial bone structure and mice, bovine, and porcine models, hampering the ability to translate biomechanical findings between species. This is further exacerbated by differences in muscle activation patterns during mastication between species. Combining the morphological differences with varying force vectors and magnitudes from mastication, it is no surprise that craniofacial strain regimes during mastication have been shown to differ across species [59,60].

Our results raise some interesting questions. First, during both chewing and suckling, axial strain magnitudes across the squamous suture surpassed those of the metopic suture, begging the question of why the squamous suture does not also close early. Comparison to our strain regimes within the metopic suture–which showed increased shear strain at regions that are clinically observed to close later—suggests that the squamous suture may be maintained patent by the high magnitude shear strains it experiences. To our knowledge, there are no data showing a causal link between cyclical forces and squamous suture closure as there is for metopic and sagittal sutures. However, there is cadaver-based evidence that increased strain across the

## Frontal and orbital maximum principal strain

**Fig 10.** Maximum principal strain in the upper face during right-sided chew (top 2 rows) and suckling (bottom 2 rows). Warmer colors indicate increased magnitude of strain. The frontal prominence, supraorbital region, and lateral orbital wall in the severe metopic craniosynostosis model (X1) experienced lower strains than the models with physiologically closed metopic sutures (C1, C2). In the non-metopic craniosynostosis models (A1, A2, B1, B2, C1, C2), partial and full closure of the metopic suture correlated with increased strains in the aforementioned regions. The less severe metopic craniosynostosis model (X2) had strain regimes that mirrored that of the physiologically closed suture models.

squamous suture arising from mastication, similar to those seen in our models, may cause higher levels of interdigitation and fractal complexity in the squamous suture, though this effect is not observed till the later decades of life [61]. Shear, thus, may be a significant modulator of suture front osteogenesis and deserves further attention.

Second, the pattern of axial strain across the metopic suture changes as the suture begins to close. Though it initially mirrored the clinically observed closure sequence when fully patent (Model A1), as the suture narrowed with intermediate closure (Model A2), the pattern became less pronounced, and eventually disappeared once closure began (Models B1, B2). This can be explained by the changes in material properties with closure, and/or mechanical strain acting as a trigger for a cascade of cell signaling between the dura and cranial vault, which may eventually become independent of the mechanical strain present initially [25,62].

## Frontal and orbital von Mises stress

**Fig 11.** Stress (von Mises) in the upper face during right-sided chew (top 2 rows) and suckling (bottom 2 rows). Warm colors indicate increased magnitude of stress, while cool colors indicate low magnitudes of stress. The frontal prominence, supraorbital region, and lateral orbital wall in the severe metopic craniosynostosis model (X1) experienced lower von Mises stress than the models with physiologically closed metopic sutures (C1, C2). The less severe metopic craniosynostosis model (X2) had stress regimes that mirrored that of the physiologically closed suture models.

Third, one might have expected a larger difference in strain and stress distributions in the C2 model given the relatively older age of 3 years and larger size of the cranium. However, compared to C1, the other fully physiologically closed metopic suture model of 7 months, minimal differences in strain and stress values across the cranial vault sutures or cranium were noted. This could be a result of the larger muscle mass in C2 being neutralized by the increased mass of the cranial vault sutures, or merely too small of a sample size to ascertain a difference.

Fourth, though prior studies have found differences between ectocranial and endocranial strain with suggested implications for which side closes first, our models did not find such differences [57,63]. There is currently no consensus in the literature as to whether suture closure begins ectocranially or endocranially. Early rat models from Moss et al. showed that chondroid tissue bridges first appear on the endocranial side, but this was refuted by Manzanares et al.,

who found that the endocranial versus ectocranial sites of initial metopic suture fusion in human cadaver studies varied across sutures [64,65]. Sahni et al. showed that lambdoid, coronal, and sagittal sutures first started closing on the endocranial side, but as their study included only adult autopsies, they were unable to examine the metopic suture [66]. The lack of difference between endocranial and ectocranial strain in our models may be a result of modeling limitations, driven by medical CT resolution and the tetrahedral element size in our models.

In addition to physiologic suture closure, our work opens up questions regarding the impact of chewing and suckling in non-physiologic states, e.g. isolated metopic craniosynostosis. Our models showed no significant difference in axial strains across the remaining cranial vault sutures between models with metopic craniosynostosis and those with physiologic metopic closure. However, relative to physiologic metopic suture closure, our model with more severe metopic craniosynostosis (X1) was observed to have decreased maximum principal strain and von Mises stress in the supra-orbital, lateral-orbital, and lateral frontal bone (Figs 8 and 9). Notably, this area with decreased strain corresponds to the frontal and supraorbital retrusion seen clinically in metopic craniosynostosis [67]. Given the evidence that cyclical forces drive bone growth, this finding suggests that physiologic chewing and suckling not only play a role in the closure of the metopic suture, but that in the setting of metopic craniosynostosis they may contribute to the development of clinically observed dysmorphic craniofacial features [68]. Interestingly, pre-mature pathologic metopic suture closure has been reported to begin in the second trimester, well before one might expect chewing and suckling to be the primary biomechanical forces at play [69,70]. However, our results support the hypothesis that physical forces play a role in physiologic closure, and that lends support to the hypotheses that physical forces experienced in-utero may play a part in early pathologic metopic suture closure, as evidence has shown that metopic craniosynostosis is not solely a result of genetics [71].

As is the case for any computational experiment, the ideal model would be validated with in-vivo measurements. However, an in-vivo model of physiologic chewing and suckling in human infants would be unethical and impractical, and so to gauge the validity of our models we turn to other methods. Including multiple finite-element models based on multiple patient scans in our study allows us to show congruent strain regimes and magnitudes between patients at similar stages of metopic suture closure. In addition, loads from chewing and suckling travel along the classically understood vertical and transverse maxillary buttresses. This implies that the cranium is loaded in a fashion, both in terms of muscle force vectors and boundary conditions, that corresponds to the current understanding of the physiologic state [40,54]. Direct comparison to prior animal models is challenging not only because of significant differences in morphology and muscle activation patterns, but because there is contradicting data on the strain across sutures during mastication. In general, animal models with similar cranial vault morphology to humans show patterns of compression and tension across the sutures that correspond to our models. Porcine studies have shown tension during the power stroke across the interfrontal suture and compression at the nasofrontal sutures, with relatively higher magnitudes of strain in the anterior interfrontal suture [32,55,72]. As we found in our human models, Behrent et al. found the sagittal suture to be placed under medial-laterally directed tension during chewing in macaques, but the findings are limited due to the use of single-element gauges [33]. The nonhuman primate for which the best in vivo data are available is the capuchin monkey *Cebus/Sapajus* from Byron et al. [73]. In vivo strains were measured at three sites along the sagittal suture in adult capuchins using rosette strain gages. The capuchin data reveal in vivo strain magnitudes similar to those modeled here during chewing and suckling. They also show that the sagittal suture is not loaded in purely laterally directed tension, as is often assumed, but is subjected to torsion or twisting, probably due to asymmetrical activity of the temporalis muscle. However, the data show that tensile

strains are larger at the front of the sagittal suture than the back, and this is associated with greater sinuosity in the anterior suture than more posteriorly [74]. Conflicting data exists for rat models, such as those reported by Shibazaki et al., who measured the interfrontal suture to be in compression during chewing [31]. Their work showed low strains at the interfrontal suture prior to its closure, and similarly low strains in the sagittal suture which remains open. As such, they concluded that strain patterns do not contribute to interfrontal suture closure. Though this may well be the case for cranial vault development in rats, the differences in strain regimes between rats and other species, both in existing literature and our results, as well as the fact that only the posterior interfrontal suture closes in rats, all suggest that caution should be exercised in using their conclusions to make definitive statements about human cranial vault development.

Our conclusions are limited by a set of factors common to finite-element analysis and computational experiments, such as a limited ability to include thin or small structures due to CT scan resolution and a relative inability to fully discern causation from correlation. Most relevant to our experimental set-up is a relatively poor characterization of the material properties of the developing cranial vault and patent sutures in existing literature, with a wide variation in the Young's modulus and Poisson's ratio cited in the literature. To account for the wide variation, we mirrored the work of Jasinoski et al., who took an average of values across several sources [51]. In addition, our sensitivity analysis of material properties showed an expected increase in strain magnitudes with a decrease in Young's modulus, but no appreciable differences in the patterns or relative distributions (S3 Fig) [52,53]. Beyond material properties, we have limited data on how jaw muscles are active during chewing and suckling by infants. The EMG activation patterns we incorporated into our models for chewing are based on adult chewing, but it is reasonable to believe that chewing during infancy may activate masticatory muscles in a different or less synchronized manner. In the case of suckling, EMG data primarily exists for the masseter and temporalis, but not for the medial pterygoid. Finally, we have not modelled intracranial pressure nor radial expansion of the developing brain, which prior studies have linked to cranial vault development [53]. As such we are unable to comment on whether a slowing pace of brain growth, in addition to or in conjunction with, masticatory forces, further contributes to cranial vault suture closure.

With the aforementioned limitations considered, to our knowledge, this work is the first to show that strain patterns that arise from the estimated physiologic forces of chewing and suckling in the developing human infant cranial vault correspond to the clinically observed timing and pattern of physiologic metopic suture closure, as well as clinically observed abnormalities in frontal bone morphology in isolated metopic craniosynostosis. Given the existing evidence that forces and their resulting strains can accelerate suture closure and stimulate osteogenesis, our work supports the theory that forces from chewing and suckling contribute to the physiologic closure of the metopic suture in humans, and in the case of trigonocephalic crania, may contribute to some of the clinically observed dysmorphic craniofacial features. Further work is necessary to determine whether the role that biomechanical forces play in physiologic metopic suture closure may also contribute to pathologic craniosynostosis, or whether such force might be manipulated in vivo to ameliorate synostosis or re-synostosis in operative cases.

## Supporting information

**S1 Table. Details of models included in study.**
(XLSX)

**S2 Table. Muscle directional vectors, cross-sectional areas, EMG activation factors, and force vectors used in finite- element models.** Negative palatal pressure of 87mmHg was

applied over surface of palate, total resulting force included here.
(XLSX)

**S1 Fig. Methodology for mid-sagittal plane theta calculations.** Relevant strain values were isolated for nodes in the sutures along the mid-sagittal plane of the cranial vault, including the nasofrontal, metopic, sagittal, and anterior fontanelle. (Panel A) The coordinate of each node was transformed into a 2-dimensional polar coordinate system with $\theta$ defined as $arctan\left(\frac{U_x}{U_y}\right)$, with $U_x$ and $U_y$ representing the x and y-coordinates of the node respectively. Note, our coordinate system has the *x* axis as superior-inferior and the *y* axis as anterior-posterior. (Panel B) Strain values were grouped and averaged by the $\theta$ value, with ranges of $\theta$ corresponding to each suture along the mid-sagittal plane. (Panel C).
(TIF)

**S2 Fig. Impact of varying material properties, specifically Young's modulus (E) and Poisson's ratio (v) on maximum principal strain regimes.** From top to bottom, material properties were: (Top) bone E = 418 MPa; v = 0.27, sutures $E$ = 50 MPa; v = 0.30 | (Middle) bone $E$ = 1300 MPa; v = 0.27, sutures $E$ = 50 MPa; v = 0.30 | (Bottom) bone $E$ = 6000 MPa; v = 0.27, sutures $E$ = 50 MPa; v = 0.30. Increasingly warm colors indicate higher levels of strain. As expected, choosing less stiff material properties (lower $E$) resulted in higher magnitudes of strain, but the pattern and distribution was relatively unchanged. For example, across the cortical bone, the highest strain values in each model were seen along the right zygomatic arch and right inferior orbital rim, and strain values in the frontal bone were highest near the frontal eminences.
(TIF)

**S3 Fig.** Sensitivity analysis comparing maximum principal strain during right-sided chewing with the foramen magnum constrained in translation along all axes (right) and unconstrained (left). Increasingly warm colors indicate higher levels of strain. Strain magnitudes and patterns across the cranium are nearly identical between the two models, with the only notable difference being increased strain in the inferior skull base sutures bordering the foramen magnum in the constrained model.
(TIF)

**S4 Fig. Sensitivity analysis comparing maximum principal strain during right-sided chewing with varying bite points during right-sided chew.** From top to bottom, the bite points are (top) 2nd molar, (middle) lateral incisor, and (bottom) 1st molar. Increasingly warm colors indicate higher levels of strain. The more anterior bite points (lateral incisor and 1st molar) show increased strain along the medial orbital floor and inferior nasal rim, but strain patterns magnitudes across the remaining cranial vault are largely similar.
(TIF)

**S5 Fig. Axial stress and strain in the superior-inferior ($\sigma_{xx}$ and $\varepsilon_{xx}$) and medial-lateral ($\sigma_{zz}$ and $\varepsilon_{zz}$) directions in each of the models during suckling.** Left panel depicts stress, right panel depicts strain. In all models, superior-inferior stress and strain were highest along the classically understood lateral maxillary vertical buttress, from the zygoma along the lateral orbital wall. The high levels of superior-inferior axial stress along the lateral orbital wall corresponded to tensile (positive, red) superior-inferior strain in the region, with corresponding compressive (negative, blue) superior-inferior strain along the medial orbital wall. Medial-lateral stress and strain were highest along the classically understood transverse maxillary buttress, specifically the lower transverse buttress along the alveolar process and the upper transverse buttress along the inferior orbital rim. Stress and strain following the classically

understood maxillary buttresses provides supporting evidence that the models are correctly modeling the physiologic mechanical landscape.
(TIF)

## Author Contributions

**Conceptualization:** Pranav N. Haravu, Callum F. Ross, Olga Panagiotopoulou, Russell R. Reid.

**Data curation:** Pranav N. Haravu, Shelby L. Nathan, Russell R. Reid.

**Formal analysis:** Pranav N. Haravu, Miguel Gonzalez, Callum F. Ross, Olga Panagiotopoulou.

**Investigation:** Pranav N. Haravu, Miguel Gonzalez, Shelby L. Nathan, Callum F. Ross, Olga Panagiotopoulou, Russell R. Reid.

**Methodology:** Pranav N. Haravu, Callum F. Ross, Olga Panagiotopoulou, Russell R. Reid.

**Resources:** Olga Panagiotopoulou.

**Software:** Olga Panagiotopoulou.

**Supervision:** Callum F. Ross, Olga Panagiotopoulou, Russell R. Reid.

**Visualization:** Pranav N. Haravu, Miguel Gonzalez, Olga Panagiotopoulou.

**Writing – original draft:** Pranav N. Haravu, Shelby L. Nathan, Callum F. Ross, Olga Panagiotopoulou, Russell R. Reid.

**Writing – review & editing:** Pranav N. Haravu, Miguel Gonzalez, Shelby L. Nathan, Callum F. Ross, Olga Panagiotopoulou, Russell R. Reid.

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
