## [Decision Letter · Decision Letter 0]

12 Apr 2023

Dear Dr. Reid,

Thank you very much for submitting your manuscript "The biomechanics of chewing and suckling in the infant: a potential mechanism for physiologic metopic suture closure" for consideration at PLOS Computational Biology. As with all papers reviewed by the journal, your manuscript was reviewed by members of the editorial board and by several independent reviewers. The reviewers appreciated the attention to an important topic. Based on the reviews, we are likely to accept this manuscript for publication, providing that you modify the manuscript according to the review recommendations.

Sincerely,

Mark Alber, Ph.D.

Section Editor

PLOS Computational Biology

Mark Alber

Section Editor

PLOS Computational Biology

Reviewer's Responses to Questions

**Comments to the Authors:**

Reviewer #1: Authors have used finite element method to quantify the level of mechanical strain across the calvarial sutures in eight individuals modelling chewing and suckling. The study has a particular focus on the metopic suture to comment on a potential mechanism for physiologic fusion of this suture (this would be challenging, not considering the brain growth). Nonetheless, to the best of my knowledge, this is indeed the first study in this field to comment on the level of strains that calvarial sutures are experiencing during infancy as a result of chewing and sucking.

Overall, I find the study novel and informative. It would be great if authors can somehow address the following points in the revised manuscript.

1) Can you please include a figure of all the eight models (A1, A2, B1, B2 etc.) so the reader can visualise the overall morphology of the sutures across the models.

2) Please outline the rough number of elements that were used to mesh the models.

3) In the limitation paragraph of the discussion please address that the radial expansion of the brain during the first year of life was not modelled in this study. And it could be that it is the dynamic between the lowering in the pace of brain growth and the rise in the level of chewing that causes fusion of the metopic suture etc.

4) Can you please comment/look into the level of principal strains in one of the models e.g. A1 to comment whether metopic suture is predominantly under tension or compression i.e. compare the 1st vs 3rd principal strain in this suture.

Specific comments:

L84-94 & L111-112 in relation to the role/impact of mechanical forces on the cranial sutures you may consider following recent studies:

Soh SH, Rafferty K, Herring S. Cyclic loading effects on craniofacial strain and sutural growth in pigs. Am J Orthod Dentofacial Orthop. 2018;154(2):270-282.

Moazen, M., Hejazi, M., Savery, D., Jones, D., Marghoub, A., Alazmani, A., & Pauws, E. Mechanical loading of cranial joints minimizes the craniofacial phenotype in Crouzon syndrome. Scientific reports,2022; 12(1), 9693.

L141 “…and Table 1. A summarized schematic of the …”

L495 “…cyclical forces drive suture closure,…” please edit this. I think there is strong body of literature to suggest that, it is the net polarity of the strain in a suture that perhaps dictate its fate i.e. patency or closure.

L519 please comment if the skull base joints were modelled in this study or not? i.e. were synchondroses modelled with soft tissues properties?

L534 “Muscle forces were estimated using….”

L561 “…measured from human unilateral chewing for the superficial…” this statement in the light of L142-143 is abit confusing i.e. were the results presented in L137 caused by bilateral contraction of the specified muscle groups or unilateral contraction.

L588 please introduce a separate section and call it “Material properties” there detail the material properties that were used on the base model which its results are described in the main text. I appreciate that these are outlined in the Sensitivity Analysis section (L606) but i think it is more appropriate to separate the two to avoid any confusion as per choice of the material properties that were used to generate the results described e.g. in L138-158.

L598 “…average nodal strain from the endo- and ectocranial surface nodes of the suture.” Can you please clarify this further i.e. were the average values reported per sutures calculated across the whole volume of the sutural elements or only their surface values (both on the endo and ectocranial).

Reviewer #2: I had the privilege to review this article submitted for publication to PLOS Computational Biology.

In this paper, finite element modelling is used to estimate suture strains during suckling and chewing in paediatric subjects aged 3 months - 3 years with different levels of metopic suture fusion.

This is an extremely well designed study and the results show promising correlation between suture closure and strain levels. The methodology is very well described and the results are presented reasonably clearly, considering the amount of information is conveyed.

I have very little to comment on an otherwise already very good manuscript.

- The authors may consider to reshuffle the paragraphs and have the methodology as second section to make the manuscript more readable (I had to read it this way to understand)

- The author should comment on the patient selection and state the reasons why the normal subjects received CTs.

- The authors should show frontal, lateral and top view of all the 8 patients, possibly in scale to appreciate also differences in size

- How were the axis oriented? I presume the frankfurt plane was assumed horizontal. The selection of the origin affects the results in figure 7, you may want to state how it was chosen.

- The authors simplify the strain analysis of the strain actoss a suture using the axial strain perpendicular to the main direction of the suture. Although this is a reasonable simplification, the authors should consider to provide some indication of the error introduced by this assumption.

- The authors should comment on some of the intra-subject variability: although a clear trend is visible, patients A1 and A2 have relatively different strain levels shown in figure 5 and figure 6. Does the age of patient C2 have an effect on the results?

- How does the current analysis relate to the pathological closure of the metopic suture in craniosynostosis: suture closure has been reported to start in the second trimester and trigonocephaly has been reported as early as at 20 weeks post-conception

(https://doi.org/10.1080/14767058.2017.1335706), can you comment on this in light of your results?

**Have the authors made all data and (if applicable) computational code underlying the findings in their manuscript fully available?**

Reviewer #1: Yes

Reviewer #2: Yes

PLOS authors have the option to publish the peer review history of their article (what does this mean?). If published, this will include your full peer review and any attached files.

Reviewer #1: No

Reviewer #2: **Yes: **Alessandro Borghi

Figure Files:

Data Requirements:

Reproducibility:

References:

---

## [Decision Letter · Decision Letter 1]

30 May 2023

Dear Dr. Reid,

We are pleased to inform you that your manuscript 'The biomechanics of chewing and suckling in the infant: a potential mechanism for physiologic metopic suture closure' has been provisionally accepted for publication in PLOS Computational Biology.

Best regards,

Mark Alber, Ph.D.

Section Editor

PLOS Computational Biology

Mark Alber

Section Editor

PLOS Computational Biology

Reviewer's Responses to Questions

**Comments to the Authors:**

Reviewer #1: Authors have addressed my original comments and i have no further comments on this paper and recommend it to be accepted. A nice study for the literature for years to come.

Reviewer #2: Thanks for addressing all the comments raised int the review. The manuscript can be accepted in the current form.

Well done!

**Have the authors made all data and (if applicable) computational code underlying the findings in their manuscript fully available?**

Reviewer #1: Yes

Reviewer #2: Yes

PLOS authors have the option to publish the peer review history of their article (what does this mean?). If published, this will include your full peer review and any attached files.

Reviewer #1: No

Reviewer #2: **Yes: **Alessandro Borghi

---

## [Editor Report · Acceptance letter]

19 Jun 2023

PCOMPBIOL-D-23-00102R1 

The biomechanics of chewing and suckling in the infant: a potential mechanism for physiologic metopic suture closure

Dear Dr Reid,

I am pleased to inform you that your manuscript has been formally accepted for publication in PLOS Computational Biology. Your manuscript is now with our production department and you will be notified of the publication date in due course.

With kind regards,

Jazmin Toth
